# Resin Composites in Posterior Teeth: Clinical Performance and Direct Restorative Techniques

**DOI:** 10.3390/dj10120222

**Published:** 2022-11-27

**Authors:** Lucas Pizzolotto, Rafael R. Moraes

**Affiliations:** 1Private Practice, Porto Alegre 90460-210, Brazil; 2School of Dentistry, Universidade Federal de Pelotas, Rua Gonçalves Chaves 457, Pelotas 96015-560, Brazil

**Keywords:** restorative dentistry, operative dentistry, adhesive dentistry, incremental technique, bulk-fill composites, shade-matching composites, dental anatomy

## Abstract

Resin composites are the most versatile restorative materials used in dentistry and the first choice for restoring posterior teeth. This article reviews aspects that influence the clinical performance of composite restorations and addresses clinically relevant issues regarding different direct techniques for restoring posterior teeth that could be performed in varied clinical situations. The article discusses the results of long-term clinical trials with resin composites and the materials available in the market for posterior restorations. The importance of photoactivation is presented, including aspects concerning the improvement of the efficiency of light-curing procedures. With regard to the restorative techniques, the article addresses key elements and occlusion levels for restoring Class I and Class II cavities, in addition to restorative strategies using different shades/opacities of resin composites in incremental techniques, restorations using bulk-fill composites, and shade-matching composites.

## 1. Introduction

Resin composites are the most versatile restorative materials used in dentistry and the first choice for restoring posterior teeth. Unlike dental amalgam, resin composites allow for the use of minimally invasive restorative procedures. Amalgam was an important material for restoring the posterior dentition in the past, especially because of its low technique sensitivity and long-term durability [1]. However, up-to-date adhesive practices use additive procedures in direct, semi-direct, and indirect versions of restorations for a variety of clinical conditions with little space for amalgam [2].

The first generations of resin composites had limitations such as low wear resistance and high polymerization shrinkage [3]. Several modifications in the formulation of resin composites were made over time to overcome clinical issues and extend the durability of restorations. These changes included the use of dimethacrylate monomers with higher molecular masses and lower polymerization stress, the incrementation of the inorganic filler loading and reduction of the filler particle size, the improvement of the resin matrix–filler particle interaction, and the use of more efficient photoinitiator systems [4,5].

Concurrently, new discoveries were made regarding the knowledge of the use of bonding systems in addition to the development of new restorative techniques, accessory materials, and devices [2]. The publication of clinical studies with long follow-up times [6,7,8,9,10,11,12] was fundamental to consolidate resin composites as major restorative materials. Although the selection of materials and the appropriate use of restorative techniques are important aspects of the clinical durability of restorations, factors related to the professional and patient need to be considered to understand the durability of resin composites in the mouth, especially with respect to patients’ risks [13,14].

Although the use of resin composites for direct restorations may be considered a consolidated clinical option in dentistry [2,10,11], novel materials are constantly being developed and new restorative strategies are being proposed. In addition, the literature is always being updated with new clinical studies; thus, it is worthwhile to review and provide a fresh perspective on this topic. The aim of this article is to provide an overview of the aspects that influence the clinical performance of resin composite restorations and address clinically relevant issues regarding different direct techniques and resin composites for restoring posterior teeth that can be performed in various clinical situations.

## 2. Materials and Methods

PubMed, Scopus, and Web of Science electronic databases were searched for articles investigating the clinical performance of direct resin composite restorations placed in posterior teeth. The search strategy used a combination of keywords: dental, composite, restoration, longevity, durability, and performance. Articles were additionally searched manually in the grey literature using the Google Scholar database. All clinical studies with posterior direct composites were eligible but special attention was given to articles published in the last 10 years and investigating variables influencing the performance of posterior composites. Laboratory and practice-based studies were also considered for inclusion if relevant to the scope of the article.

## 3. Results and Discussion

An overview of the main clinical aspects that influence the performance of posterior restorations is presented and discussed in the following subsections, in addition to resin composite materials and various direct techniques that can be used for restoring posterior teeth.

### 3.1. Factors Influencing the Clinical Performance of Restorations

Table 1 summarizes the main aspects that may influence the longevity of composite restorations in posterior teeth as reported in long-term clinical studies [15,16,17,18,19,20,21,22,23,24,25,26,27,28,29,30,31,32,33,34,35,36,37,38]. Although failures are expected to occur in any type of dental restoration, resin composites can be considered the materials of choice for posterior teeth [2,10,11]. Compared with ceramics and amalgam, for example, composites enable the use of strictly additive techniques and, thus, facilitate the greater preservation of dental structure. When composite restorations fail, the prognosis of recovery of the remaining dental structure tends to be favorable. Finite element analyses indicated that large dental cavities restored with a composite showed better biomechanical behavior compared to amalgam [39].

The limitations of resin composite restorations involve the development of wear, marginal defects, surface staining, and marginal pigmentation over the course of their clinical service [40]. The failures of composite restorations are mainly related to the occurrence of fractures and adjacent caries [2,10,11,37]. However, the majority of clinical failures of composites allow for repair procedures. Repairs are minimally invasive procedures that have been shown to extend the clinical durability of resin composites in both posterior and anterior teeth [41,42]. A significant current challenge in restorative dentistry is the rehabilitation of patients with bruxism, clenching, and parafunctional habits as these challenges may cause mechanical overloading of restorations and propagate wear, fractures, and failures [43]. Since patients with severely worn teeth may require many interventions and repairs to their dental restorations with time, an international consensus assessed that additive techniques are preferable in these cases [44]. More invasive restorative techniques such as ceramic crowns should not be considered as a first clinical option for posterior teeth. It should also be kept in mind that ceramics will depend on resin-based materials for bonding to the dental structures.

### 3.2. Resin Composites for Posterior Restorations

Different brands of resin composites are launched on the market every year with promises to deliver excellent performance. Manufacturers often introduce updated versions of ‘old’ materials claiming improved clinical behavior, but these updates are usually associated with new packages or new brand logos, and certainly with increased costs. A recent 5-year clinical study did not find any difference in the clinical performance between an updated compared with an old version of the same resin composite [45]. A recent clinical study on posterior composite restorations that was followed up for 33 years was carried out using materials available in the 1980s [7]. A study with a 20-year follow-up showed that three microhybrid composites could effectively be considered universal composites as they have presented satisfactory clinical performance in both anterior and posterior teeth [16].

With the exception of some low-quality materials that may eventually be available in the market, as well as resin composites that are exclusive for use in anterior restorations (e.g., microfills), most composites have sufficient physical–chemical properties for use in the posterior dentition. Clinical studies with bulk-fill materials have shown clinical performance similar to conventional composites after up to 10 years [46,47]. Recently, shade-matching composites have been introduced to facilitate restorative procedures by eliminating the need for a shade selection step [48]. These composites have the chromatic ability to match tooth shades from A1 to D4 with a single shade, but the topic warrants investigation as no long-term clinical studies have been published so far.

Regarding the cost-effectiveness of restorations, it has been suggested that all resin composite alternatives are likely to be inferior to amalgam [49]. Another investigation concluded that amalgams were more cost-effective than composites even for replacing existing Class II amalgam restorations [50]. However, an in vitro study indicated that low-cost resin composites showed physical–chemical properties comparable to a composite from a well-known, consolidated manufacturer [51]. The authors also indicated that in a situation of scarce resources, low-cost composites could be an acceptable cost-effective alternative. In any case, the clinical decision making should consider factors beyond the cost-effectiveness of materials, including the patients’ values.

Table 2 presents a summary of the composites available for restorations in posterior teeth and their main characteristics. Each material type and brand presents its virtues and limitations, and longevity will not solely depend on selecting a specific material. Thus, dentists might choose which composite to use based on the characteristics that they consider optimal for clinical usage. The selection of resin composites for posterior teeth should consider their handling characteristics, ease of use, and the availability of shades and pigments, in addition to other criteria that may influence their clinical use. An interesting study conducted a ‘blind test’ with dentists who were requested to rate different composites presented to them in generic, standardized packages [52]. In a second round, the same materials were presented in their original packaging to the dentists. The materials from more ‘famous’ brands and with more attractive packaging received poorer scores in the blind round but had higher scores in the second evaluation. This finding suggests that several factors may influence dentists’ perceptions of the materials that they use.

### 3.3. The Importance of Photoactivation

Photoactivation is an essential step for ensuring the longevity and success of direct composite restorations because the conversion of monomers into polymers depends exclusively on light irradiance. However, most photopolymerization studies are based on laboratory experiments and scarce clinical evidence on the topic is available. Monowave and polywave light-curing units are available in the market. Monowave LEDs emit light that is concentrated in a narrow wavelength range, usually centered around an absorption peak of the photoinitiator camphorquinone (operating in the blue region). Polywave or multi-wave units have at least two different LEDs and thus emit light in the blue region and other wavelength ranges such as violet or ultraviolet regions, thus enabling the polymerization of materials containing photoinitiators other than camphorquinone [53]. In vitro studies indicate that both types of units work well [54]. However, the preliminary evidence suggests that polywave LEDs are less likely to suffer a significant reduction in irradiance over time via constant use [55], a finding that could be related to factors other than their ability to emit light at different wavelengths. A summary of the important aspects concerning light-curing units is described as follows, in addition to information regarding methods to improve the efficiency of photoactivation procedures [56,57,58,59,60].

#### 3.3.1. General Aspects Regarding LED Light-Curing Units

Give preference to light-curing units with irradiance levels of ≥1000 mW/cm^2^, with large light guide tips that may avoid light concentration in small cavity areas, and units with a shape and/or design that makes them suitable for accessing the posterior dentition;The use of polywave units is not mandatory unless the resin composite manufacturer indicates otherwise;Routinely test the irradiance level of the unit to ensure that there has been no loss of power with clinical usage and aging;Cordless units should not be used when their battery is low (<25%);Tilting, inclination, or distancing of the light guide from the cavity may generate shadow areas or reduce the irradiance reaching the composite increment;Avoid several sequential photoactivations, e.g., irradiance times rounding off 2 min or more, and use intervals because the repetition of photoactivations may reduce the irradiance level;Do not purchase products of uncertain origin, brands without regulatory clearance in your country, or those for which the manufacturer does not provide auditable information about product specifications and tests.

#### 3.3.2. Aspects to Improve the Efficiency of Photoactivation Procedures

Learn how and train yourself to deliver maximum levels of light irradiance to the composite increment by correctly positioning the tip parallel and as close as possible to the irradiated surface.Keep the light guide’s tip clean, use barriers such as transparent polyvinyl chloride film without creating areas where the barrier becomes thick or poorly adapted.Learn the photoactivation time of each resin composite that you use; they normally require energy doses between 6 to 24 J/cm^2^, which are calculated by the product of the irradiance multiplied by the exposure time (e.g., 1000 mW/cm^2^ × 20 s = 20 J/cm^2^).When using darker composite shades, increase photoactivation times by at least 10 s.When the tip’s adaptation to the cavity is not optimal, or the photoactivation is performed at an inclination or at a greater distance from the composite, the exposure time should be extended.Do not reduce the photoactivation time recommended by the manufacturer, even when using light-curing units with irradiance above 1500 mW/cm^2^.Do not expect that a large cavity should be light-cured using only one central exposure because there is a chance that the light will not adequately reach and polymerize all cavity margins.When necessary, use overlapping photoactivation procedures to reach larger areas, especially if the cavity creates shadowy areas during light irradiance.In deep cavities with low dentin thickness, be careful not to overheat the structure, especially when photoactivating the adhesive and the first composite increments. If necessary, use an interval before a new photopolymerization procedure and/or use an air stream to cool down the dental structure.Cover soft tissues with gauze to avoid exposure to light, especially when the irradiance is high, and the procedure is performed near the dental cervical margin for long times.The dentist and assistant should use visual protection (orange goggles) so they can observe the correct positioning of the tip throughout the photopolymerization procedure and to avoid indirect exposure to the blue light. If possible, patients should also use orange goggles for protection.

### 3.4. Key Elements and Occlusion Levels in Posterior Restorations

Understanding three key elements improves dentists’ perception and facilitates the reconstruction of structures including those that have suffered major structural damage such as cusp losses or smaller proportions such as Class I or II cavities [61]. The first element is the external contour or dental silhouette, which reveals the individuality of each posterior tooth through the embrasures. This feature is responsible for the appearance that the teeth are not united in the arch and for the external perimeter characteristic of the first lower molar and its five cusps, for instance. The second element is the occlusal table or anatomical occlusal surface, delimited by the circumscribed area between the cusp tips. It is related to the antagonist teeth and the vertical dimension of occlusion. We find the grinding slopes on the occlusal table, which are the principal actors of most of the masticatory process and food crushing.

The third element is the occlusion map, which reveals the diagram within the occlusal table. The map is specific for each posterior tooth and will guide the restoration from the conformation of the resin composite increment with respect to dentin until the end of the enamel reconstruction. The occlusion map establishes numerous restoration parameters, such as the appropriate volume of the cusps, the position of slopes, the inclination of edges, the correct positioning of the central groove, and the fossae and pits. With respect to the correct drawing of the occlusion map, the delimitation of the structures and their relationship becomes more predictable, thereby reducing the number of occlusal adjustments performed after the end of the sculpture. Hence, for the restorative process to become easy and fast, the dentists must train their perception before using their motor capacity. Several methods can be used to memorize and understand the anatomical drawings formed by different occlusion maps.

Figure 1 shows the stylization of the occlusion maps using the ‘Y technique’ for the posterior teeth, thus allowing the primary anatomy to be restored. The Y technique is a form of occlusal design memorization that uses a symbolic pattern to locate and map anatomical structures. The letter Y is inserted in different positions, dividing the tooth into its proper parts, arranging the anatomy in a stylized manner so that all the structures are positioned and the restoration is performed more efficiently. Although there are several anatomical variations among the same types of teeth, the occlusion map design hardly changes.

Another parameter of fundamental importance for posterior restorations constitutes the so-called four occlusion levels, elevations, and depressions revealed by the dental anatomy. As shown in Figure 2, the cusp tips form the first and highest level, followed by marginal ridges (second level), the main sulcus (third level), and pits and fossae, which form the fourth occlusion level. When restoring a tooth without regarding this height relationship, an extensive occlusal adjustment may be necessary at the end. If the central groove is constructed above the marginal ridge level, for example, the restoration will not have the appropriate height (it will be too ‘high’), requiring adjustment until reaching the occlusal levels.

### 3.5. Restorative Techniques

#### 3.5.1. Restoration Considering the Shades and Opacities of Composites

The variety of resin composites and techniques used to restore posterior teeth is enormous and seems to grow every year. For conventional composites, we can mention some techniques based on the number of shades used (Figure 3): monochromatic (one shade), bichromatic (two shades), or polychromatic (more than two shades).

**Monochromatic technique**: This is used mainly in shallow cavities. A single increment of the composite can be used for chromatic enamel (composites that present a shade such as A1 and A2). Body composites or universal composites can also be used as they present an intermediate shade and degree of translucency, that is, neither low translucency as composites for dentin nor translucency as composites for enamel. Non-VITA shades of enamel composites are an additional option because they have interesting optical characteristics with which toemulate the dental enamel.**Bichromatic technique**: This technique is widespread and frequently used in posterior direct restorations. It can be applied in several cavity configurations that present a structural loss of dentin and enamel, thus employing two shades/opacities of a composite. For the reconstruction of the dentin portion, a composite with lower translucency and high saturation (such as dentin A3.5 or A4) is a good option. Remember the need to maintain a space of approximately 1.5 mm for the reconstruction of the next enamel portion. The measurement of the spaces can be performed with instruments with rounded tips, e.g., instruments with two balls with diameters of 1.5 mm and 2.5 mm. The professional experience acquired in the clinical routine also helps. A less saturated and higher value enamel or body/universal composite (enamel A1 or A2 or body/universal) should be used for the enamel portion.**Polychromatic technique**: This technique uses more than two shades/opacities of the composite, aiming at an excellent degree of mimicry of the dental structures, but requires the dentist to dedicate more attention and time to performing the layering process. The reconstruction is initiated from the dentin portion using the same principles of the bichromatic technique concerning shade (composites for dentin of high saturation and low translucency), with the difference being the need to maintain a slightly larger space (approximately 2.5 mm) for two layers of enamel to be accommodated above. Next, the reconstruction of the chromatic enamel layer is performed, usually using composite A1 or A2, and again paying attention to the maintenance of space for the final layer of approximately 1.5 mm. As a final covering layer, achromatic composites are the ideal materials. Achromatic enamel resin composites are those that have no shade themselves. They exhibit optical properties very similar to natural enamel and help form a semi-translucent milky layer. These composites are value modulators; color value is an optical property of great importance in restorative dentistry. Unlike chromatic resin composites, achromatic composites generally have no relation to the VITA shade scale, nor do they follow the same nomenclatures among manufacturers.

#### 3.5.2. Bichromatic Technique for Class I Cavities

The bichromatic technique is widely used because it offers the restoration of dentin and enamel objectively and distinctly by using a specific type of resin composite for each tissue. Several methods of performing the bichromatic technique are available. Here, we address the technique that achieves speed, predictability, and effectiveness in the restorative treatment. Cavities that will receive direct restorations should have all unsupported enamel removed and the margins well-finished, thereby favoring an intimate relationship with the adhesive system and restorative material. Be conservative when removing unsupported enamel and excavating carious dentin; the preparation should be minimally invasive, preserving the remaining structure as much as possible. Selective caries removal is preferable; creating a bevel at the cavosurface angle may or may not be performed—this is left to the professional’s discretion.

Figure 4 presents a clinical case regarding the replacement of an amalgam restoration (Figure 4a) with a composite. Conditioning with 37% phosphoric acid was performed only on the enamel (selective enamel-etching technique) for 30 s using an acid etchant with good thixotropy (Figure 4b), which facilitates the control of the etched area. After washing the tooth with air/water spray (minimum 30 s), the adhesive system was applied to the dry tooth. In the example, a two-step self-etch adhesive was used (Figure 4c). In cavities with sharp internal angles or an irregular pulpal wall (lunar crater appearance), a flowable composite can be applied to even out the pulpal wall, seal the dentin surface, and improve the contact of the regular resin, which will be inserted later, with the internal angles of the cavity. The flowable composite should not be left exposed at the cavity margin. Some cavities restored with amalgam may present pulpal floors stained by amalgam corrosion products or reactional dentin, generating a darkened aspect on the dentin. In these cases, an opacifier liner is an option to improve the aesthetic aspect of the restoration (Figure 4d). The reconstruction of the dentin portion can be performed using high-saturation and low-translucency composites. In the technique presented here, the dentin was reconstructed in a single composite increment, and its volume was controlled to leave space for the last enamel layer. The dentin composite increment is brought into the cavity in a singular fashion. Soon after, the occlusion map is drawn using a very fine instrument with gentle but constant movements (Figure 4e). This occlusion map will guide the preparation of the covering layer (enamel), giving more predictability to the position of the structures and preventing future occlusal adjustments.

Soon after constructing the artificial dentin, optical characterizations can be performed at the discretion of the professional and patient using light-cured tints (Figure 4f). The purpose of the tints is to increase the perception of low and high regions of the tooth and the three-dimensionality of the restoration, as well as to emulate the pigmentation of central grooves and white spots that may affect the enamel, allowing the restoration to appear more natural. The intensity of characterizations should be based on the neighboring teeth and/or with respect to the restored tooth. An example is the teeth of elderly patients, which may exhibit more optically loaded features such as pigmented ridges. In contrast, young people’s teeth may not exhibit such features as frequently. An A1 or A2 shade of enamel composite with regular consistency but lower viscosity can be used for the enamel layer (covering). When used, the less viscous regular composite enables more anatomical details to be created when worked cusp by cusp (Figure 4g). Conventionalviscosity composites keep the sculpture in position longer than low-viscosity composites and show excellent results when used for single increment enamel construction and the simultaneous sculpting of all structures. The formation of the central groove of the restoration is achieved by the light contact of each increment of the resin composite that will be accommodating and occupying its proper space. It is suggested not to use an instrument to groove and demarcate the composite; this hinders the procedure and increases the clinical time, besides creating irregularity in the resin composite. When a cusp-by-cusp restorative technique with resin composite increments is used, each increment must be photoactivated as soon as each structure is rebuilt.The immediate final aspect of the restoration is shown after the removal of the rubber dam and occlusal adjustment (Figure 4h).

Figure 5 details the restorative procedure of a Class I cavity (Figure 5a) using a single increment for dentin reconstruction (Figure 5b). The occlusion map is initiated using a very fine instrument whose tip touches the bottom of the cavity (Figure 5c). This procedure allows the increments to be separated (Figure 5d,e) and speeds up dentin construction. This separation considers the cavity configuration factor (C-factor) by reducing problems related to the bonding of opposing walls that could aggravate the consequences of the polymerization stresses. Thus, this technique can be considered a variation of the incremental technique. In this proposal, the construction of the dentin in a single step allows for clinical time gain, including the time taken for the photoactivation of increments.

#### 3.5.3. Technique for Class II Restorations

Perhaps the most significant challenges in Class II restorations are re-establishing a good contact point and achieving the optimal sealing of the proximal gingival wall margins. Pre-curved partial matrix systems are excellent for achieving an effective contact point. Some of these are commercially available as kits, such as Pallodent V3 (Dentsply), Composi-Tight 3D XR (Garrison), Unimatrix (TDV), andmyClip 2.0 (Polydentia/Quinelato), each with its own differentials. Most systems have pre-curved partial matrices of various sizes for different situations and teeth. Moreover, most of them include wedges (wooden, plastic, or elastic), rings (of different sizes and with fitting devices for premolars and molars), and forceps for the insertion of the rings. Some systems have their own forceps; others are compatible with conventional clamp holders, whilesome do not need forceps.

The clinical steps for Class II restorations are presented in Figure 6. Each item has its function in the restorative accessory kit. The rings are primarily used to perform light dental separation, which is fundamental to obtaining an effective contact point. The rings also help in achieving the correct contour of the proximal ridges. The wedges are intended to increase the intimate contact and adaptation of the matrix with the tooth at the gingival margins, which are critical regions. Thus, they prevent the formation of gaps and excesses and keep the matrix in position. The adhesive system is applied after the matrix and wedges were positioned. A recent laboratory study evaluated Class II composite restorations prepared when the adhesive was applied before or after the application of the interproximal matrix and wedge [62]. The authors observed that both scenarios may have disadvantages: when bonding was carried out before the matrix’s positioning, the adhesive was spread on tooth areas well beyond the restoration borders; when bonding was performed after the matrix’s positioning, the adhesive was embedded in the composite’s surface. The suggestion in both cases was to carefully finish the cervical margins to remove an adhesive-rich layer and improve marginal adaptation.

Wedges are also used to aid tooth separation in order to obtain a contact point and should be customized, when necessary, for each situation. This can be executed with scalpel blades or sandpaper discs so as to not interfere in the space destined to be the contact point. Different matrix sizes should also be considered so that they are positioned 1 mm above the marginal ridge of the neighboring tooth. Another maneuver that may be important in achieving success in this type of procedure is to lightly burnish the inserted matrix to generate passive contact with the neighboring tooth. Once the entire matrix system has been stabilized and its steps properly evaluated, the layering process can begin.

#### 3.5.4. Restorations Using Bulk-Fill Composites

Bulk-fill composites can assist the professional in the restorative steps by simplifying and accelerating the layering process in posterior teeth. This type of composite can be used in restorative strategies that enable the insertion of increments thicker than 2 mm, as previously discussed. In cases where the cavity respects the depth limits recommended for such materials, the use of a single increment or increments combined in different ways may be an alternative. Figure 7 presents different restorative possibilities combining bulk-fill materials (flowable or regular consistency) with conventional composites. These strategies can be employed for Class I or Class II restorations and consider different cavity depths.

Flowable bulk-fill resin composites may represent a good option for the rapid reconstruction of the dentin portion of the restoration. The flowable form should not remain as the last layer, overlay layer, or restorative margins because of its lower wear resistance due to the lower volume of filler particles compared to regular bulk-fill materials. The ‘bulk and body’ technique [63] (Figure 8a–e) involves using flowable bulk-fill resin for dentin reconstruction, and, as the last layer, a conventional composite is employed to provide adequate resistance to masticatory loads. A regular bulk-fill resin composite could also be used for the topping. Due to its greater translucency, the bulk-fill composites are more greyish than conventional composites. For this reason, in the bulk and body technique, the conventional composite for enamel reconstruction compensates for the absorption of light by the bulk-fill composite, lending a better appearance to the restoration and preventing it from becoming greyish.

Regular bulk-fill resin composites can be used in deep cavities, to reduce the number of increments, or as a single increment when the cavity height does not exceed the maximum increment size limit recommended by the manufacturer, as is the case for the ‘bulk and go’ technique [63] (Figure 8f–h). Some manufacturers indicate that flowable composites can be exposed to the oral environment, including the cavity margins. However, there is still uncertainty whether the lower amount of filler could lead to the lower stability of the margins and thus interfere with the longevity of restorations.

#### 3.5.5. Restoration Using Shade-Matching Resin Composites

The clinical technique using a shade-matching resin composite for a Class I restoration is shown in Figure 9. The restorative steps are virtually the same as those used for regular composites except for the absence of shade selection. Clinicians may still use separate dentin and enamel layers for a better characterization of the tooth anatomy. Tints can be used to improve the 3D aspect of restorations.

## 4. Conclusions

Dental resin composites are versatile materials that can be used in various direct clinical techniques to restore posterior teeth, including different shades and opacities of composites (monochromatic, bichromatic, and polychromatic techniques), incremental filling techniques, regular and flowable bulk-fill materials, and shade-matching composites. The clinical performance of restorations depends on a number of factors including variables related to the restored tooth, the restorative materials and techniques employed, the patient’s risks, and professional clinical decisions.

## Figures and Tables

**Figure 1 dentistry-10-00222-f001:**
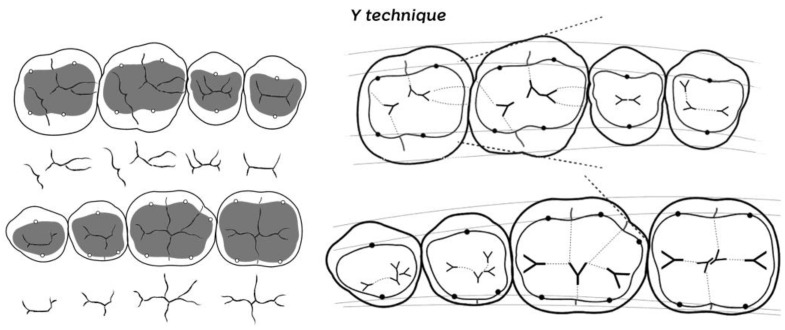
Stylization of the occlusion maps using the Y technique for posterior teeth. Left: The three key elements used to construct the posterior teeth. The occlusion maps are inserted into the occlusal tables (gray areas), which in turn have their boundaries established by the cusp tips (white spheres) surrounded more externally by the external tooth silhouette or contour. Right: A stylized drawing of the occlusion maps of the posterior teeth using the Y technique.

**Figure 2 dentistry-10-00222-f002:**
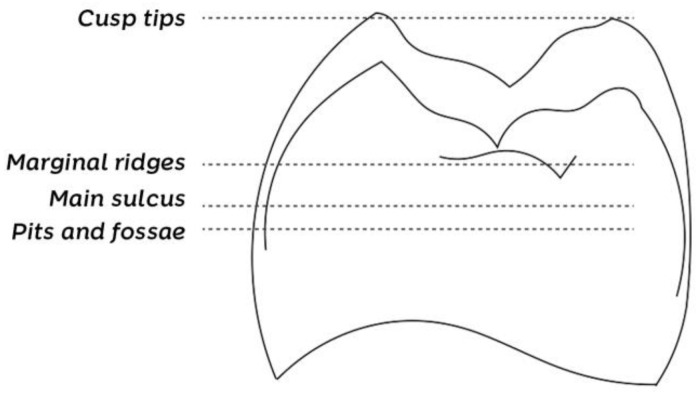
Four occlusal levels that must be considered when preparing posterior restorations: cusp tips, marginal ridges, main sulcus, and pits and fossae. Failure to respect this height relationship of the anatomical structures may necessitate extensive occlusal adjustment at the end of the restoration.

**Figure 3 dentistry-10-00222-f003:**
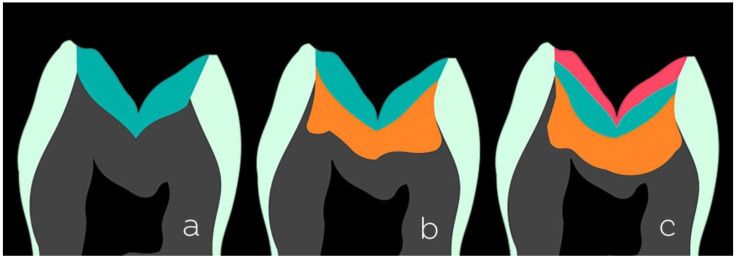
Different restorative techniques using conventional resin composites based on the number of shades used (represented by dark green, orange, and red colors): monochromatic (**a**), bichromatic (**b**), and polychromatic (**c**).

**Figure 4 dentistry-10-00222-f004:**
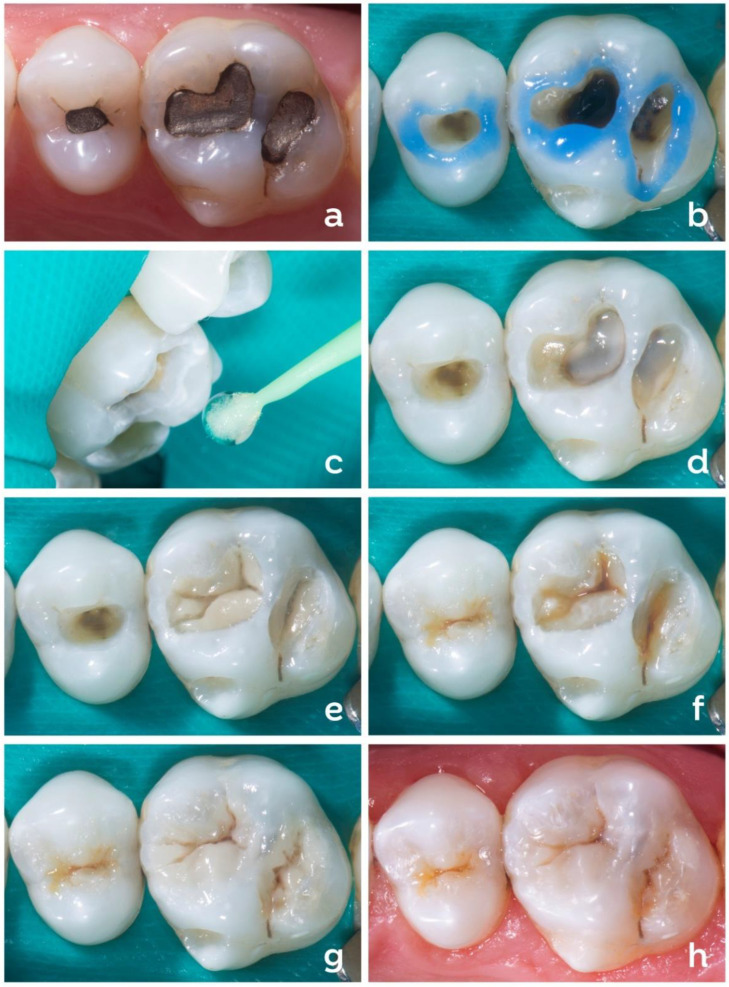
Initial aspect of an amalgam restoration to be replaced (**a**). Selective enamel etching with phosphoric acid (**b**). Application of self-etch adhesive system (**c**). A thin layer of opacifier is applied on darkened dentin (**d**). Insertion of dentin composite and design of occlusal map (**e**). Ochre, brown, and white tints applied to increase three-dimensionality (**f**). Enamel composite inserted using a single increment (**g**). Immediate final aspect of the restoration after removal of rubber dam and occlusal adjustment (**h**).

**Figure 5 dentistry-10-00222-f005:**
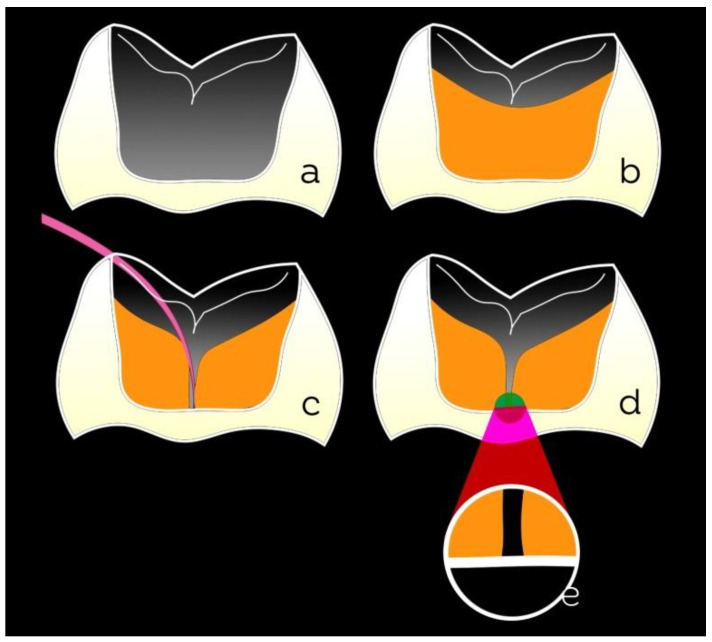
Restorative technique using single increment for dentin reconstruction. The cavity (**a**) is filled by single increment insertion of the dentin composite without photoactivation (**b**). The occlusion map is started using a very fine instrument (**c**). The instrument’s tip touches the bottom of the cavity, which allows the increments to be separated (**d**,**e**) and speeds up dentin construction.

**Figure 6 dentistry-10-00222-f006:**
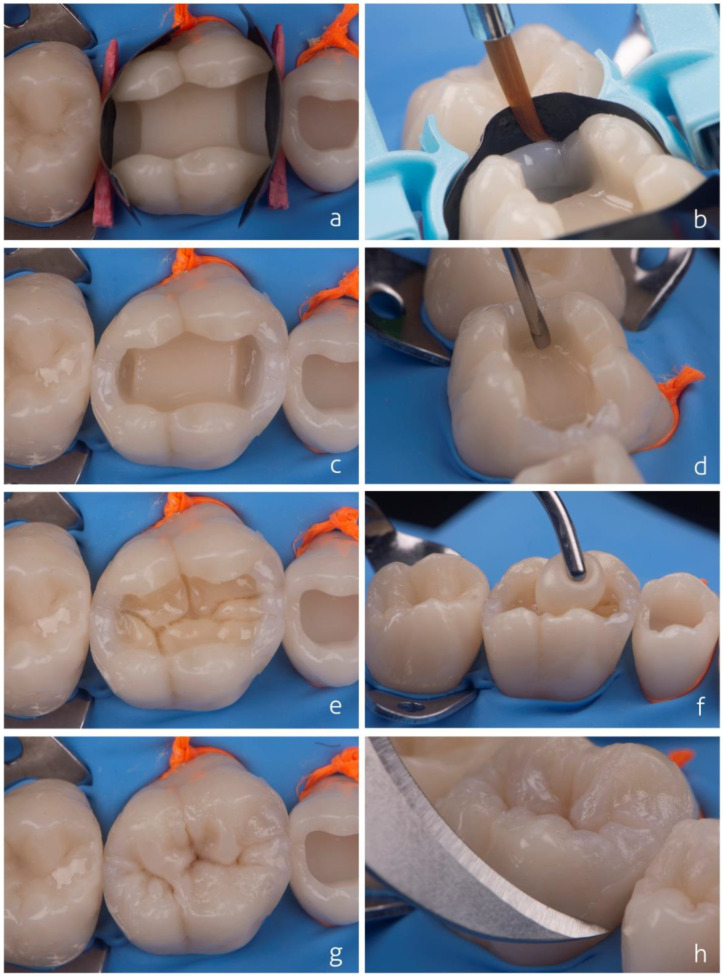
Class II restoration with composite. Positioning of partial matrices and wedges before application of adhesive system (**a**). Composite is placed in the proximal area and pre-polished with a brush before photoactivation (**b**). Aspect of the cavity with both proximal walls built with enamel composite (**c**). Proximal box filled with flowable bulk-fill composite (**d**). Restoration of artificial dentin and occlusal map design (**e**). Insertion of enamel composite on the occlusal surface (**f**). Aspect of the final occlusal anatomy (**g**) and finishing of proximal surfaces with scalpel blade (**h**).

**Figure 7 dentistry-10-00222-f007:**
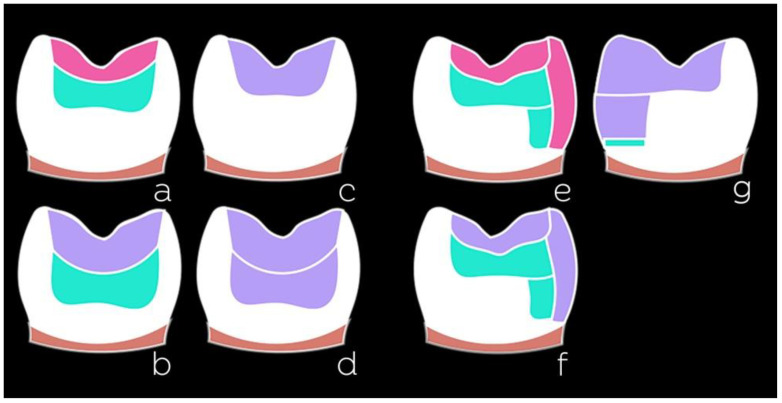
Different restorative strategies using bulk-fill resin composites. Green = flowable bulk-fill composite; Pink = conventional composite; Purple = regular bulk-fill composite. Reconstruction of the dentin portion with flowable bulk-fill composite and a conventional composite as the last layer (**a**). Reconstruction of the dentin portion with flowable bulk-fill composite and regular bulk-fill composite as the last layer for deeper cavities (**b**). Cavities that respect the limit of the composite polymerization depth can be restored with a single increment of regular bulk-fill composite (**c**). Deeper cavities can be restored with two increments of regular bulk-fill composite (**d**). Class II cavities can be restored with conventional composite emulating the enamel and its interior filled with flowable bulk-fill composite for dentin reconstruction. Similar to example (**e**), a regular bulk-fill composite can be used for the enamel (**f**). It is also possible to use flowable bulk-fill composite as a liner on the gingival wall for all techniques described here. Two increments of a larger volume of regular bulk-fill composite added above a liner with flowable bulk-fill composite (**g**).

**Figure 8 dentistry-10-00222-f008:**
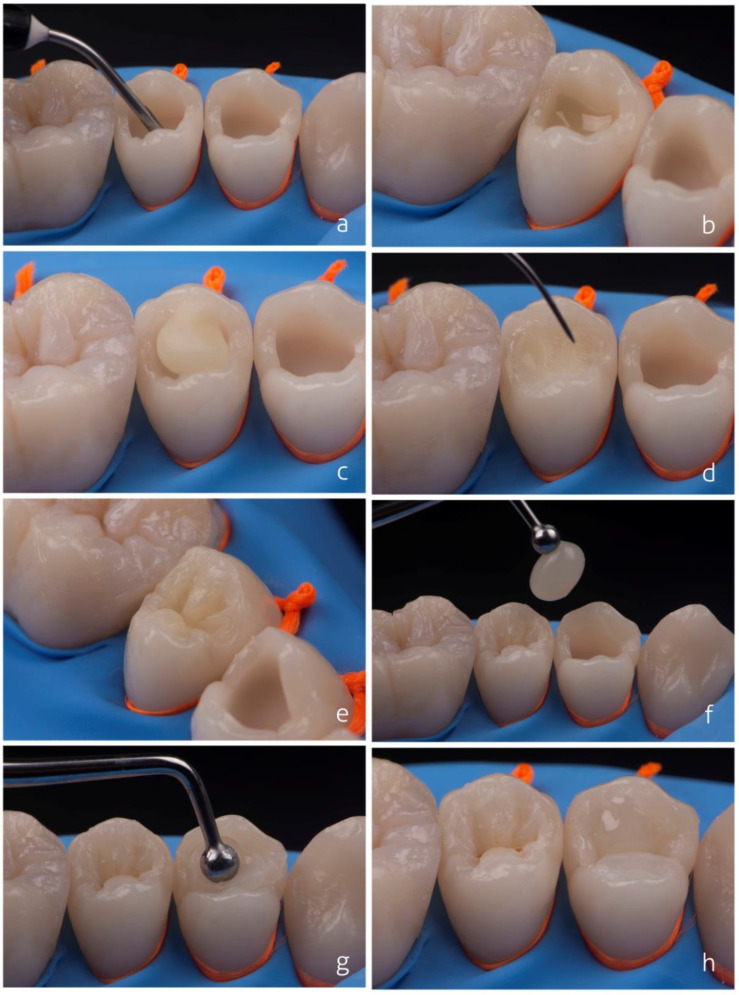
Examples of ‘bulk and body’ and ‘bulk and go’ restorative techniques. Bulk and body: restoring the dentin portion of the cavity with flowable bulk-fill composite (**a**,**b**), insertion of enamel composite (**c**), sculpture (**d**), and final aspect (**e**). Bulk and go: insertion of regular bulk-fill composite as single increment (**f**), accommodation of the increment (**g**), and final aspect of the sculpture (**h**).

**Figure 9 dentistry-10-00222-f009:**
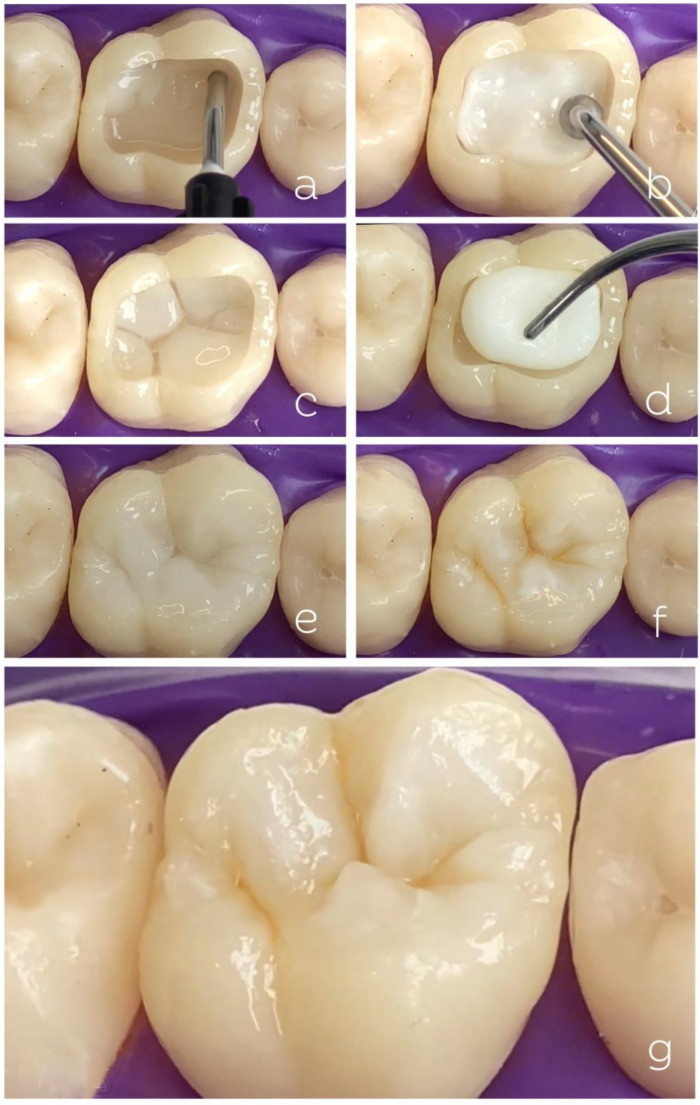
Class I restoration using a shade-matching resin composite. Filling irregular areas and rounding internal angles with shade-matching flowable composite (**a**). Insertion of shade-matching composite to replace dentin (**b**). Aspect of the artificial dentin after occlusal mapping and design (**c**). Insertion of the same shade-matching composite for the preparation of dental enamel in a single increment (**d**). Final aspect of artificial enamel carved in single increment (**e**). Extrinsic characterization with ochre, brown tints, and dentin composite for bleached teeth emulating regions of large amount of enamel as main lobes of the inclined cuspal planes (**f**). Final aspect of the restoration (**g**).

**Table 1 dentistry-10-00222-t001:** Factors that have significant or limited influences on the clinical longevity of direct resin composite restorations.

**Significant Influence on Longevity**	
Cavity size and volume	Greater loss of dental structure and cavity walls increases the risk of failures [15]
Location of tooth in dental arch	Restorations in molars fail more often than in premolars, posterior restorations fail mainly due to fracture and secondary caries, and esthetic failures are prevalent in anterior restorations [16]
Cervical extension	Deep restorative margins increase the risk of failures [17]
Enamel etching	When enamel is present, etching with phosphoric acid is recommended [18]
Selective removal of carious tissue	Selective removal reduces the chances of pulp complications compared to more aggressive caries excavation methods [19]
Endodontic treatment	Restorations in endodontically treated teeth fail more often than in vital teeth [20]
Use of thick cavity liners	Glass-ionomer cement layers should be thin (<1 mm); the use of calcium hydroxide cement may not be necessary in deep cavities [21]
Presence of adjacent teeth	Restorations fail more often when no adjacent teeth are present or when the restored tooth is the last in the dental arch [22,23]
Patient’s age and sex	Studies generally report higher risk of restoration failures in men, children, and elderly patients [24,25]
Patient’s risks	Risks related to new caries lesions, occlusal stress, periodontal health, radiotherapy, smoking, dietary habits, and parafunctional habits (e.g., nail biting) increase the chances of failures [13,14,26,27]
Frequent change of dentists	Changing dentists increases the risk of unnecessary interventions and restoration failures [28]
**Limited Influence on Longevity**	
Restorative technique	It is important for the dentist to select a technique that will lead to fewer mistakes, to use few composite increments, to ensure low internal porosity, and provide optimal marginal adaptation and sealing [29,30,31]
Adhesive system	Simplified adhesives were usually regarded as having more failures than 3-step etch-and-rinse and 2-step self-etch adhesives, but current evidence points out that no significant differences may exist among different bonding systems [32,33,34]
Isolation method	When isolation of the operative field is carried out properly, no significant differences in the long term are observed between restorations carried out using a rubber dam or cotton rolls [35]
Beveling of enamel margins	It does not seem to affect the longevity of restorations and could be used at the dentist’s discretion [36]
Resin composite type and brand	The restorative technique and patient-related factors are more important to the longevity of restorations than material brands or types [7]
Polymerization shrinkage of composites	Contemporary composites have shrinkage levels compatible with long-lasting restorative procedures and techniques [37]
Finishing and polishing system	Although important for the quality of restorations, there is still insufficient evidence on the effect of different polishing systems for the longevity of restorations
Marginal staining	Marginal pigmentation does not entail marginal or secondary caries. Updated caries diagnosis methods and fewer unnecessary interventions will increase the longevity of restorations [7,38]

**Table 2 dentistry-10-00222-t002:** Characteristics of resin composites used for posterior restorations.

Technique *	Type of Resin Composite *	Characteristics
Conventional incremental technique (increments typically up to 2 mm in thickness)	Microhybrid/Nanohybrid	A resin matrix of dimethacrylate monomers filled with two types/sizes of inorganic particles: nanofillers (<100 nm) and microfillers (>1 μm)
	Submicron/Supranano	Particle size is above the nano-scale (>100 nm) but below the micro-scale (<1 μm), typically 0.2–0.4 μm
	Nanofill	The filler system contains only discrete nanoparticles or nano-agglomerates
Bulk-fill technique (increments may be larger than 2 mm in thickness)	Flowable	A fluid resin matrix of dimethacrylate monomers with low polymerization shrinkage, whose filler particles can be larger than those in conventional composites. The increased translucency allows for photopolymerization of thicker increments. Flowable bulk-fills should be used as a restorative base, which needs to be covered by a final, top layer of conventional composite
	Regular	Similar in composition to the flowable composite but with higher viscosity. There is no need for a top layer of conventional composite. Typical increments are up to 4 mm in thickness
	Mixed	Composites with a regular viscosity that may be rendered more fluid through application of sonic vibration. Increments may be up to 5 mm in thickness
Other nomenclatures for resin composites	Universal	Composites that can be used to restore both anterior and posterior teeth. More recently, the nomenclature has also been used for shade-matching composites
	Single shade or shade-matching	Composites with chromatic technology for color matching different tooth shades (e.g., from A1 to D4) by using a single resin composite shade

* Nomenclatures may vary among manufacturers.

## Data Availability

Not applicable.

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
