# Peer review of "Resin Composites in Posterior Teeth: Clinical Performance and Direct Restorative Techniques"

_dentistry, 2022, doi:10.3390/dj10120222_

Round 1

Reviewer 1 Report

Dear Authors, 

you made a great work! However, some improvements are mandatory before acceptance. 

Author Response

Thank you for your time reviewing our manuscript. Below please find responses to points you raised about aspects that should be improved and/or edited in the manuscript. Changes in the text were highlighted using blue-colored font.

  • “Although the selection of materials and appropriate use of restorative techniques are important aspects for the clinical durability of restorations, factors related to the professional and the patient need to be considered for understanding the durability of resin composites in the mouth, especially patients’ risks [12,13].” I think it is also interesting to consider the possibility of using resins for other reasons, such as the distribution of masticatory forces that are very different from other materials, or the characteristic of shock absorption at the implant level, as indicated by:

“Reda, R.; Zanza, A.; Galli, M.; De Biase, A.; Testarelli, L.; Di Nardo, D. Applications and Clinical Behavior of BioHPP in Prosthetic Dentistry: A Short Review. J. Compos. Sci. 2022, 6, 90. https://doi.org/10.3390/jcs6030090”

R: The distribution of stresses arising from masticatory forces when composite is compared to amalgam was mentioned in subheading 3.1, as requested. A reference was inserted. The mention to amalgam is also in line with a comment from another reviewer. The suggested article by Reda et al. addresses BioHPP, described as a partially crystalline poly ether ether ketone (PEEK) strengthened by using ceramic. In our view this paper investigated a topic not really related to the scope of our article.

In the materials and methods section:

  • “Is it possible to insert the exact search string in the materials and methods? What if some Journal for manual searching have also been selected?”

R: This comment was really helpful. We improved the description of search methods. We did not provide an exact search string because we did not use a pre-defined, unique combination of keywords. Your comment also was helpful for reminding us to provide more details to the readers about the nature of the review and how articles were searched. Thank you again.

Results are easy to understand and comprehensive. All the studied characteristics were reported in tables which are clear and concise. Please check the copyright of the explanatory figures inserted in the manuscript.

R: The copyright of all figures used in the manuscript belong to the authors.

Reviewer 2 Report

Congratulations for this article! Paper is well written and a significant contribution to the literature. No improvement needed by my side. 

Author Response

Thank you for your time reviewing our manuscript.

Reviewer 3 Report

it is an interesting study. COnsidering it is "just"  a review of the literature, i would add few aspects. First of all, i would increase the references, i feel it is too weak from that point of view. The introduction is a little short. 

I feel that a comparison with the amalgam would be interesting, considering the adavantages and disadvantages for each material. Furthermore i would coinsider in what kind of patients the composites are not the best option. I would express also the weak aspect of these materials. 

Finally i would spend few sentences about the costs and the access to these materials. 

Thank You

Author Response

Thank you for your time reviewing our manuscript. Below please find responses to points you raised about aspects that should be improved and/or edited in the manuscript. Changes in the text were highlighted using blue-colored font.

It is an interesting study. considering it is "just"  a review of the literature, I would add few aspects. First of all, I would increase the references, I feel it is too weak from that point of view. The introduction is a little short.

R: We did not find the original article “weak” in quoting references, we used 53 references throughout the text. However, the revised manuscript was improved in terms of referencing, which is also in line with comments from another reviewer. The revised manuscript cited 10 new references. In addition, the Intro chapter was improved as suggested.

I feel that a comparison with the amalgam would be interesting, considering the adavantages and disadvantages for each material. Furthermore I would coinsider in what kind of patients the composites are not the best option. I would express also the weak aspect of these materials.

R: These comments were accepted and the manuscript revised. We mentioned amalgam in the Intro section but a longer assessment of amalgam would be beyond the scope of this article. We also have mentioned the limitations of composite restorations (subheading 3.1).

Finally I would spend few sentences about the costs and the access to these materials.

R: We payed attention to this topic in the revised manuscript. Three new studies regarding cost-effectiveness were cited (subheading 3.2).

Reviewer 4 Report

Dear Authors,

generally, the write-up of the paper is good. 

The pictures are amazing, and the documentation is top-level.

Here are some suggestions to improve the manuscript.

The authors should cite and support the sentences better. Most of the sentences do not have a reference and shall be considered the personal opinion, which is incorrect for a scientific paper.

Line 22-44:

The authors should consider extending the consideration of composites' evolution to anterior restorations.

The authors could consider adding a sentence like the following:

“Furthermore, the evolution of resin-based composites has also influenced anterior restorations allowing highly esthetic results and predictable outcomes”. The authors could cite both PART I (PMID: 25975063) and PART II (PMID: 25223143) of the article: “Stratification in anterior teeth using one dentine shade and a predefined thickness of enamel: a new concept in composite layering”.

Line 60-2:

The authors wrote:

“Although failures are expected to occur in any type of dental restoration, resin composites can be considered the materials of choice for posterior teeth. “

Please add a reference to this sentence.

The authors could consider using Reference 36 (Opdam NJ, et al.)

Line 96-7:

please a reference to support the introduction of shade-matching composites 

Lines 116-180:

The article refers to “Resin composites in posterior teeth: Clinical performance and direct restorative techniques”

The reviewer thinks that composite characteristics and modeling techniques are beyond the purpose of this article. Adding also a 2-page section on photoactivation makes this paper closer to a book chapter than to an article.

The authors could consider removing the photoactivation part (From line 116 to Line 180).

Line 367-375:

Please specify if the adhesive system has to be applied before or after the matrix.

The authors could consider citing a paper that outlines differences in the adhesive application (before or after the matrix). The article (that the authors could cite) is: https://doi.org/10.1016/j.jdent.2020.103494

Lines 411:

Please add a scientific reference on the “‘bulk-and-body’ technique”. The reviewer searched the common databases without retrieving any paper. If a technique is cited it should be published on an indexed paper.

Lines 422:

Please add a scientific reference on the “‘bulk-and-go’ technique”. The reviewer searched the common databases without retrieving any paper. If a technique is cited it should be published on an indexed paper. A technique cited in the literature (and therefore citable) is for example the simultaneous modeling technique.

Author Response

Thank you for your time reviewing our manuscript. Below please find responses to points you raised about aspects that should be improved and/or edited in the manuscript. Changes in the text were highlighted using blue-colored font.

The authors should cite and support the sentences better. Most of the sentences do not have a reference and shall be considered the personal opinion, which is incorrect for a scientific paper.

R: We have added more references to the manuscript to support some of the statements. This comment is in line with those from another reviewer.

Line 22-44: The authors should consider extending the consideration of composites' evolution to anterior restorations. The authors could consider adding a sentence like the following: “Furthermore, the evolution of resin-based composites has also influenced anterior restorations allowing highly esthetic results and predictable outcomes”. The authors could cite both PART I (PMID: 25975063) and PART II (PMID: 25223143) of the article: “Stratification in anterior teeth using one dentine shade and a predefined thickness of enamel: a new concept in composite layering”.

R: This comment regarded the first paragraph of the manuscript, which was revised to include some information about amalgam. This was done because two reviewers indicated that amalgams should be mentioned because the paper concentrates in posterior restorations. In this context, we found difficult to bring up anterior restorations in the Intro chapter.

Line 60-2: The authors wrote: “Although failures are expected to occur in any type of dental restoration, resin composites can be considered the materials of choice for posterior teeth.“ Please add a reference to this sentence. The authors could consider using Reference 36 (Opdam NJ, et al.)

R: We added references to that passage.

Line 96-7: please a reference to support the introduction of shade-matching composites.

R: Done.

Lines 116-180: The article refers to “Resin composites in posterior teeth: Clinical performance and direct restorative techniques” The reviewer thinks that composite characteristics and modeling techniques are beyond the purpose of this article. Adding also a 2-page section on photoactivation makes this paper closer to a book chapter than to an article. The authors could consider removing the photoactivation part (From line 116 to Line 180).

R: We do agree with you that the manuscript is a bit long in some aspects and it could be seen as a boon chapter – which is naturally a narrative review with educational purposes. This article has a similar goal. We believe that the content of this manuscript could be very helpful to students and clinicians regarding materials, methods, and techniques available to restore posterior teeth. Photoactivation is an essential step for preparing adequate restorations that can last long in the clinical setup. Therefore, we believe that keeping this content will make the article richer.

Line 367-375: Please specify if the adhesive system has to be applied before or after the matrix. The authors could consider citing a paper that outlines differences in the adhesive application (before or after the matrix). The article (that the authors could cite) is: https://doi.org/10.1016/j.jdent.2020.103494

R: Thank you so much for this excellent suggestion. The topic was addressed and the manuscript was quoted.

Lines 411: Please add a scientific reference on the “‘bulk-and-body’ technique”. The reviewer searched the common databases without retrieving any paper. If a technique is cited it should be published on an indexed paper.

Lines 422: Please add a scientific reference on the “‘bulk-and-go’ technique”. The reviewer searched the common databases without retrieving any paper. If a technique is cited it should be published on an indexed paper. A technique cited in the literature (and therefore citable) is for example the simultaneous modeling technique.

R: The source was indeed missing, thanks for noticing this. The reference was quoted in the revised manuscript.

Round 2

Reviewer 4 Report

Almost all the comments and requests have been addressed but the reviewer still thinks that a reference to the huge versatility of the composite use both for anteriors and posteriors shall be highlighted. Rather than referring to amalgam (in 2022 ???).

I suggest again to consider the previous comment:

Line 22-44: The authors should consider extending the consideration of composites' evolution to anterior restorations. The authors could consider adding a sentence like the following: “Furthermore, the evolution of resin-based composites has also influenced anterior restorations allowing highly esthetic results and predictable outcomes”. The authors could cite both PART I (PMID: 25975063) and PART II (PMID: 25223143) of the article: “Stratification in anterior teeth using one dentine shade and a predefined thickness of enamel: a new concept in composite layering”.

Author Response

Thanks again for your time in reviewing our manuscript. We agree that mention to amalgam was not really necessary in this manuscript, but it seems that this material is able to attract a lot of attention from dental practitioners still nowadays. Two other reviewers asked for amalgam to be addressed in the first review round. Now we added the passage you suggested about anterior restorations to the first paragraph of the Intro (highlighted in blue) and quoted both references you indicated. In our view this sentence allowed to take a bit the focus on amalgam and move the narrative again to composites. Thanks for your suggestion.